# Hexogen Coating Kinetics with Polyurethane-Based Hydroxyl-Terminated Polybutadiene (HTPB) Using Infrared Spectroscopy

**DOI:** 10.3390/polym14061184

**Published:** 2022-03-16

**Authors:** Heri Budi Wibowo, Hamonangan Rekso Diputro Sitompul, Rika Suwana Budi, Kendra Hartaya, Luthfia Hajar Abdillah, Retno Ardianingsih, Ratih Sanggra Murti Wibowo

**Affiliations:** 1Research Organization of Aeronautics and Space—National Research and Innovation Agency (BRIN), Jakarta 13220, Indonesia; hamonanganrs@gmail.com (H.R.D.S.); sbudirika@yahoo.com (R.S.B.); kendra19838@yahoo.co.id (K.H.); luhaabdillah@gmail.com (L.H.A.); re_ardian@yahoo.com (R.A.); 2Faculty of Engineering, Universitas Gadjah Mada, Jl. Grafika No. 2 Kampus UGM, Yogyakarta 55281, Indonesia; mratihh@gmail.com

**Keywords:** kinetic, polyurethane, hexogen, coating

## Abstract

The kinetics of hexogen coating with polyurethane-based hydroxyl-terminated polybutadiene (HTPB) using infrared spectrometry was investigated. The kinetics model was evaluated through reaction steps: (1) hydroxyl and isocyanate to produce urethane, (2) urethane and isocyanate to produce allophanate, and (3) nitro and isocyanate to produce diazene oxide and carbon dioxide. HTPB, ethyl acetate, TDI (toluene diisocyanate), and hexogen were mixed for 60 min at 40 °C. The sample was withdrawn and analyzed with infrared spectroscopy every ten minutes at reference wavelengths of 2270 (the specific absorption for isocyanate groups) and 1768 cm^−1^ (the specific absorption for N=N groups). The solvent was vaporized; then, the coated hexogen was cured in the oven for 7 days at 60 °C. The effect of temperature on the coating kinetics was studied by adjusting the reaction temperature at 40, 50, and 60 °C. This procedure was repeated with IPDI (isophorone diisocyanate) as a curing agent. The reaction rate constant, k_3_, was calculated from an independent graphic based on increasing diazene oxide concentration every ten minutes. The reaction rate constants, k_1_ and k_2_, were numerically calculated using the Newton–Raphson and Runge–Kutta methods based on decreasing isocyanate concentrations. The activation energy of those steps was 1178, 1021, and 912 kJ mole^−1^. The reaction rate of hexogen coating with IPDI was slightly faster than with TDI.

## 1. Introduction

Hexogen is a highly explosive material. Hexogen is widely used as an additive to increase the combustion energy of rocket composite propellants [1,2]. The modern composite propellant has at least ammonium perchlorate (AP) as an oxidizer, aluminum as solid fuel, and hydroxyl-terminated polybutadiene (HTPB) as a binder. HTPB is a polybutadiene resin copolymerized with a diisocyanate compound to form a polyurethane binder. Hexogen has a highly mechanical sensitivity and is incompatible with HTPB binders; thus, the composite propellant manufacturing process becomes unsafe [3,4,5]. In order to reduce its sensitivity and agglomerations in the manufacturing of propellants [6,7,8], hexogen has been coated with polyurethane. However, the reported thickness of the polyurethane layer is still thick, twice the original size. Polyurethane is a non-energetic binder with low combustion energy. Therefore, it needs to be used as little as possible in propellant formulations [9,10]. As layer thickness is linear to the polyurethane concentration, the polyurethane layer should be as thin as possible.

Coating kinetics is needed to determine the coated concentration and depth of coating. Hexogen is an aromatic compound with three nitro groups. Meanwhile, polyurethane is a polymer compound with a chain of urethane groups, and HTPB is a polybutadiene compound consisting of a butadiene group chain with a hydroxyl group at the end of the chain. Coated hexogen with polyurethane occurs due to the reaction of the nitro group of hexogen with the isocyanate group of polyurethane to form a diazene oxide group [6,7]. Polyurethane itself is produced by the reaction of di-alcohol with diisocyanate. The isocyanate group can react with the urethane group to form an allophanate group during the polymerization process. Thus, there will be a competitive reaction of isocyanates with diols to form polyurethanes, polyurethanes to form allophanates, and nitro to form diazene oxides. The reaction kinetics model must include all the reaction processes that occur.

The reaction kinetics of polyurethane-based HTPB and TDI has been widely studied based on the reaction between their functional groups. Many studies report the kinetics of polyurethane formation in bulk condition based on the change of polymer product properties such as viscosity [11,12,13,14,15,16], polymer weight [12,13], torque [17,18], thermal properties [15,18,19,20], and infrared spectroscopy [19,20,21]. Moreover, polyurethane formation in bulk and solution conditions has also been reported from the analysis of characteristic functional group changes based on infrared absorption data [22,23], viscosity change (rheology) [11,12,13,14,15,16], magnetic resonance/NMR [24,25], and heat flux [18,26]. Further, polymerization of polyurethane formation from HTPB with IPDI has also been conducted in both bulk and solution conditions [18,20,26,27,28,29,30,31,32,33]. The polymerization consists of several steps involving two hydroxyl groups in HTPB and two isocyanate groups. The polymerization will be terminated if all functional groups have reacted to form urethane groups. Ajithkumar et al. state that urethane groups can react with isocyanate groups to form allophanate groups as a branching reaction in the presence of excessive isocyanate [15,21]. In addition, Wibowo et al. also state that the linear and branching reaction could occur simultaneously and competitively [3,21]. The polymerization kinetics model can be measured using infrared spectrometry from decreasing isocyanate groups concentration with absorptions characteristic at 2270–2276 cm^−1^. Although the infrared spectrometry method is cheap, fast, and accurate, the reaction of nitro with isocyanates to form diazene oxide compounds has never been carried out using this method. Nonetheless, studies propose that the presence of diazene oxide groups can be distinguished from nitro compounds through infrared absorption at wavelengths of 1648 and 1758–1768 cm^−1^ [34,35]. The absorption is strong enough to be a differentiator with nitro groups at the wavelength of 1300–1500 cm^−1^, which has much interference. 

This paper studied the reaction kinetics of hexogen coating with polyurethane-based HTPB using infrared spectrometry. The quantitative analysis was based on the absorbance of isocyanate and diazene oxide as reference. The diisocyanate materials used were TDI (toluene diisocyanate) and IPDI (isophorone diisocyanate). The reaction mechanism included a competitive reaction of isocyanate with a di-alcohol from HTPB, a urethane group from polyurethane, and a nitro group from a hexogen.

## 2. Materials and Methods

### 2.1. Materials

Hexogen with a particle size of approximately 100 microns was supplied by Dahana, Co., Ltd., Indonesia. Ethyl acetate p.a. was produced by Merck, and HTPB and TDI were supplied by Dalian, Co., Ltd., Dalian, China. HTPB had a hydroxyl number of 40, average functionality of 1.9, and an average molecular weight of 3000 g/mole. Meanwhile, TDI had a 2,4- to 2,6- isomers molar ratio of 80:20, and the isocyanate value was 39 mg eq/g KOH. 

### 2.2. Instrumentations

The isocyanate and diazene oxide concentrations were measured using an infrared spectrometer FTIR (Fourier transform infrared) Hitachi IR-Prestige Serial A210043 with a standard liquid cuvet. A qualitative analysis of the isocyanate and diazene oxide concentrations was carried out by plotting the measured absorbance at wavelengths of 2265–2270 cm^−1^ and 1780 cm^−1^ to their calibration curve. The coating depth was measured with the scanning electron microscope (SEM) Phenom Word ProX Desktop [36,37].

### 2.3. Procedure

The calibration curve of the isocyanates concentration was performed by dissolving 0.1 g toluene diisocyanate (TDI) in 100 mL benzene. Approximately 10 mL of the solution was collected and analyzed with FTIR using a standard 10 mL cuvet. This procedure was repeated with 0.2, 0.3, and 0.4 until 1.0 mL of TDI. The measured absorbance of spectra at the wavelength of 2276 cm^−1^ was plotted to their concentration to create the calibration curve of isocyanate concentration. The diazene oxide concentration calibration curve was performed by dissolving 0.1 g 4,4′-dimethoxyazoxybenzene in the 100 mL benzene. Approximately 10 mL of the solution was extracted and analyzed with FTIR using a standard 10 mL cuvet. This procedure was repeated with 0.2, 0.3, and 0.4 until 1.0 mL of TDI. The measured absorbance of spectra at the wavelength of 1870 cm^−1^ was plotted to their concentration to create the calibration curve of diazene oxide concentration. 

In the next step, 4 mL HTPB, 50 mL ethyl acetate, and TDI with an NCO to OH molar ratio of 1:1 (RNCO/OH) were poured in the 500 mL beaker glass. Two hundred grams of hexogen were added and mixed for 60 min at 40 °C. Approximately 1 mL of the sample was collected and dissolved in 100 mL of benzene every ten minutes. Ten milliliters of the sample solution was poured into the cuvet and analyzed with infrared spectroscopy. The infrared absorption peaks at wavelengths of 2270 cm^−1^ (the specific absorption for isocyanate groups) [38,39,40] and 1768 cm^−1^ (the specific absorption for N=N groups) [34,35] were taken as a reference. The slurry was stirred vigorously, and then the ethyl acetate was removed by vacuum drying at 50 °C for one hour. The coated hexogen was cured in the oven for 7 days at 60 °C. The coated hexogen was sieved with a 100 micron sieve. This particle size and the coating depth were measured with SEM using the watershed segmentation method to separate particles that stick or were close together [41]. 

The temperature effect on the coating kinetics was studied by adjusting the reaction temperature at 40, 50, and 60 °C. This procedure was repeated with IPDI as a curing agent. 

The absorption of infrared spectra at a wavelength of 2267 cm^−1^ was plotted to an isocyanate calibration curve to meet the isocyanate concentration. Meanwhile, the absorption of infrared spectra at a wavelength of 1870 cm^−1^ was plotted to a diazene oxide calibration curve to meet the diazene oxide concentration. 

### 2.4. Kinetics Model

In the coating of hexogen with polyurethane-based HTPB, the hexogen with nitro functional groups was bonded with isocyanate groups from TDI and polyurethane [34,35]. Polyurethane was produced by copolymerization of HTPB and TDI [3,11,19]. The HTPB and TDI could copolymerize through the reaction of hydroxyl groups and isocyanate groups to produce linear urethane groups as illustrated in Figure 1. Every urethane compound contained two active functional groups (i.e., isocyanate or hydroxyl) that reacted to produce new urethane groups. The urethane groups would grow to produce longer urethane chains. The urethane group also reacted with a diisocyanate to produce new linear urethane groups or branching allophanate groups. This polymerization occurred by the linear or branching reaction. The copolymerization kinetics of HTPB and TDI were simplified by the reaction of their functional groups (i.e., hydroxyl and isocyanate) because the functional group reactivity was not affected by their molecular size [3]. The newest copolymerization kinetics approach of HTPB and TDI proposed a linear reaction of hydroxyl functional groups and isocyanate functional groups to produce urethane groups (Equation (1)) and a branching reaction of urethane functional groups and isocyanate groups to produce allophanate groups (Equation (2)) [3,17,42,43]. The reaction of HTPB and TDI occurred through linear and branch bond formation [3,11,15,23]. 

The kinetics model based on the hydroxyl groups from HTPB (A) reacting with isocyanate groups from TDI (B) produced urethane groups (D) via linear bonding. The urethane groups (D) then reacted with other isocyanate groups (B) to produce allophanate groups (E) via branch bonding. The reactions were expressed in Equations (1) and (2). The isocyanate groups (B) also reacted with a nitro group (F) in hexogen to produce diazene oxide groups (G) and carbon dioxide (H) [29]. The reaction was expressed in Equation (3). In the coating of hexogen, there were competitive reactions involving the isocyanate with hydroxyl groups of HTPB, urethane groups of polyurethane, and nitro groups of the hexogen.
−NCO + −OH → −NHCOO-(1)
−NCO + −NHCOO− → −N(COO−)(CONH−)(2)
−NCO + −NO_2_ → −NNO− +CO_2_(3)

## 3. Results and Discussion

### 3.1. Coated Hexogen Identification

The change in particle size and shape parameters indicated the coated hexogen as illustrated in Figure 2. The average particle size of hexogen decreased from 127.68 to 96.85 microns (hexogen PU-coated with IPDI) and 64.08 microns (hexogen PU-coated with TDI) refers to Table 1. The homogenization effect due to the dispersion in ethyl acetate solution could explain this size reduction phenomenon in which the hexogen particle breaks down in size before reacting with coating substances [44,45]. Besides the particle size, the spherical shape increased from 0.78 to 0.87 (circularity) and from 0.71 to 0.78 (roundness) using IPDI as isocyanates in PU coating. In contrast, the particle size reduction was more significant using TDI. The shape parameters, such as circularity were unchanged, yet the roundness and aspect ratio was worse than the uncoated particle. These results were similar to the hexogen coating with HTPB-based polyurethane carried out by Neudorfl et al. [34,35]. The coated hexogen was more spherical than pure hexogen. Circularity is a shape descriptor that could mathematically imply the level of similarity to a perfect circle. A circularity score of 1.0 classified a perfect circle. As the circularity value reached 0.0, the shape was an increasing number much less circular. Roundness was similar to circularity. However, it was insensitive to abnormal borders alongside the fringe of the foramen. Roundness also considers the substantial axis of the exceptional suit ellipse [46].

Infrared spectrometry further identified the polyurethane coating as shown in Figure 3. The figure overlays the infrared spectra of ethyl acetate, HTPB, TDI, hexogen, polyurethane, and coated hexogen. Coated hexogen was free from ethyl acetate as indicated by the absence of absorption characteristics of the tridueteromethyl and dideuteromethylene at 2300 and 2000 cm^−1^. The C=O stretching vibration showed progressive displacement at lower frequency with increasing deuteration. The polyurethane formed had both isocyanate and hydroxyl group absorption at 2270 and 3400 cm^−1^ wavelengths. The concentration of the hydroxyl group was difficult to measure, because it had an extensive peak angle quantitatively. Hexogen had infrared characteristic absorption to nitro groups at 1359–1328 cm^−1^ from the symmetric stretch of the para-nitro group and at 1562–1535 cm^−1^ assigned to the asymmetric stretch ortho-nitro group [34,35]. The reaction of hexogen with polyurethane produced a diazene oxide group that gave a powerful specific absorption at a wavelength of 2848 and 1758 cm^−1^ [34,35]. The adsorption of isocyanate and diazene oxide can be used to study the kinetic of hexogen coating. 

### 3.2. The Measurement of Isocyanate and Diazene Oxide Concentration 

The concentration of isocyanate solution was analyzed by plotting the infrared absorbance at a wavelength of 2276 cm^−1^ and extrapolation to the isocyanate calibration curve. Quantitative infrared spectroscopy analysis applied a standard liquid cuvet sampler for the Prestige FTIR diameter of 1 cm. The conversion of the absorbance to the concentration used the Beer law, in which the spectra absorbance is linear to their concentration if the other parameters are constant. Figure 4 illustrates the isocyanate calibration curve, where the absorbance value was linear to their isocyanate concentration and was expressed by the equation Abs = 0.1726 CNCO − 0.0053 with an *R*² value = 0.9986. These linear relationships showed no significant interference and could be used for quantitative analysis.

The diazene solution concentration was analyzed by plotting the infrared absorbance at a wavelength of 1768 cm^−1^ and extrapolation to the diazene calibration curve. Quantitative infrared spectroscopy analysis used a standard liquid cuvet sampler for the Prestige FTIR diameter of 1 cm. The conversion of the absorbance to the concentration used the Beer law, where the spectra’s absorbance was linear to their concentration if the other parameters were constant. The diazene calibration curve is illustrated in Figure 5, and shows that the absorbance value was linear to their diazene concentration and is expressed by equation Abs = 16.066 CNCO − 0.1247 with an *R*² value = 0.9973. 

### 3.3. Kinetic Evaluation

The coating kinetics of hexogen and polyurethane-based HTPB were evaluated based on the infrared absorbance at 2276 (specific absorption for isocyanate group) [15,21] and 1758 cm^−1^ (specific absorption for diazene oxide group) as a reference [34,35]. The absorbance of the sample was fitted to the calibration curve and converted to their concentrations. In order to study the kinetic model, the isocyanate and diazene oxide concentrations were plotted to the time reaction as seen in Figure 6. The decreasing isocyanate concentration meant that HTPB reacted with TDI, while the increasing diazene concentration indicated the reaction between hexogen and polyurethane. 

The kinetics model was evaluated through reaction steps including (1) hydroxyl group (A) and isocyanate group (B) to produce urethane group (D), (2) urethane group (D) and isocyanate group (B) to produce allophanate group (E), and (3) nitro group (F) and isocyanate group (B) to produce diazene oxide group (G) and carbon dioxide (H). The carbon dioxide (gas) was removed from the reactor. Hence, the reactions can be shown in Equations (4)–(6) with reaction rate constants k_1_, k_2_, and k_3_.
A + B → D(4)
D + B → E(5)
F + B → G(6)

Each step was assumed to fit the first-order reaction with the rate constant of k_1_, k_2_, and k_3_. The reactivity of the functional groups in solution did not depend on their molecular size. The decreasing concentration rate of A was expressed by Equation (7), while the decreasing concentration rates of B, D, and F were shown in Equations (8)–(10) based on the increasing concentration of G.
−dC_A_/dt = k_1_C_A_C_B_(7)
−dC_B_/dt = k_1_C_A_C_B_ + k_2_C_D_C_B_ − k_3_C_G_(8)
−dC_D_/dt = −k_1_C_A_C_B_ + k_2_C_D_C_B_(9)
−dC_F_/dt = dC_G_/dt = k_3_C_G_(10)

The diazene oxide formation rate in Equation (10) was independent of the other equations. The integration of Equation (10) with initial C_Go_ = 0 at t = 0 gave relation C_G_ to time (t) in Equation (11). The diazene oxide formation rate constant, k_3_, was calculated from the ln CG versus t curve slope as shown in Figure 7. The calculated k_3_ was 9.11 × 10^−4^ L mole^−1^·min^−1^.
ln C_G_ = k_3_t     or C_G_ = e^k^_3_^t^(11)

Equations (12) and (13) show the mole balance of hydroxyl groups (A) and isocyanate groups (B) at a specific reaction time, where C_Ao_ and C_Bo_ are the initial concentration of A and B, respectively.
C_Ao_ = C_A_ + C_D_ + C_E_(12)
C_Bo_ = C_B_ + C_D_ + 2C_E_ + C_G_(13)

Substitution of Equation (12) into Equation (13) can derive C_E_ and C_D_ as a function of C_A_ and C_B_ such as the Equations (14) and (15): C_E_ = C_Bo_ − C_Ao_ + C_A_ − C_B_ − C_G_(14)
C_D_ = 2C_Ao_ − C_Bo_ − 2C_A_ + C_B_ + C_G_(15)

Equations (11), (14), and (15) were substituted into Equation (8) to give the decrease of B as a function of C_A_, C_B_, and C_G_: −dC_B_/dt = k_1_C_A_C_B_ + k_2_C_B_(2C_Ao_ − C_Bo_ − 2C_A_ + C_B_ + e^k^_3_^t^)−k_3_e^k^_3_^t^(16)

There are two reaction rate equations (i.e., Equations (7) and (16)) with the reaction rate constants k_1_ and k_2_. The initials of C_A_ and C_B_ at t = 0 were C_A_o and C_B_o. The data source was the isocyanate groups’ concentration every ten minutes as seen in Table 2. Equations (7) and (16) were simultaneously nonlinear differential equations. There are two differential equations with three unknown variables (i.e., C_A_, k_1_, and k_2_); thus, the unknown variable can be numerically calculated or estimated using the Newton–Raphson approach. The trial variable was k_1_. Initially, k_1_ was set to 0.00001 following the value reaction rate constant of polyurethane formation based on the HTPB and TDI [15]. 

Reaction rate equations had the initial isocyanate and hydroxyl groups concentration (t = 0), namely, C_Ao_ and C_Bo_. The isocyanates concentration data every ten minutes were expressed by $C_{B}^{d}{}_{i}$, where i was time. Equations (7) and (16) were numerically solved with the Newton–Raphson method. Equations (7) and (16) were converted to the nonlinear equations of F_1_(k_1_,k_2_) and F_2_(k_1_,k_2_) with the estimated variables k_1_ and k_2_ following Equations (17) and (18).
F_1_(k_1_,k_2_) = dC_A_/dt(17)
F_2_(k_1_,k_2_) = dC_B_/dt(18)

Initially, variables k_1_ and k_2_ were set, then the new k_1_ and k_2_ were estimated by Equations (19) and (20), with n being the iteration increment.
(19)$$k_{1_{n+1}}=k_{1_{n}}+\frac{-F_{1}\left(k_{1_{n}},k_{2_{n}}\right)\frac{\partial F_{2}}{\partial k_{2}}+F_{2}\left(k_{1_{n}},k_{2_{n}}\right)\frac{\partial F_{1}}{\partial k_{2}}}{\frac{\partial F_{1}}{\partial k_{1}}\frac{\partial F_{2}}{\partial k_{2}}+\frac{\partial F_{2}}{\partial k_{1}}\frac{\partial F_{1}}{\partial k_{2}}}$$
(20)$$k_{2_{n+1}}=k_{2_{n}}+\frac{-F_{1}\left(k_{1_{n}},k_{2_{n}}\right)\frac{\partial F_{2}}{\partial k_{1}}+F_{2}\left(k_{1_{n}},k_{2_{n}}\right)\frac{\partial F_{1}}{\partial k_{1}}}{\frac{\partial F_{1}}{\partial k_{1}}\frac{\partial F_{2}}{\partial k_{2}}+\frac{\partial F_{2}}{\partial k_{1}}\frac{\partial F_{1}}{\partial k_{2}}}$$

The k_1_ and k_2_ were continuously iterated until a tolerance boundary error was achieved. The tolerance boundary error was achieved by the least squares error (SSE) and expressed in Equation (21).
(21)$$\overset{10}{\underset{i=1}{\sum }}{\left|C_{B}^{d}{}_{10i}-C_{B}^{d}{}_{10i}\right|}^{2}\leq tol$$

The value of $\frac{\partial F_{1}}{\partial x}$, $\frac{\partial F_{2}}{\partial x}$, $\frac{\partial F_{1}}{\partial y}$, and $\frac{\partial F_{2}}{\partial y}$ was approached by the function value difference from $k_{1_{n}}+\in$ to $k_{1_{n}}$, where ε → 0, expressed by Equations (22)–(25).
(22)$$\frac{\partial F_{1}}{\partial k_{2}}=\frac{F_{1}\left(k_{1_{n}},k_{2_{n}}+\in \right)-F_{1}\left(k_{1_{n}},k_{2_{n}}\right)}{\in }$$
(23)$$\frac{\partial F_{1}}{\partial k_{1}}=\frac{F_{1}\left(k_{1_{n}}+\in ,k_{2_{n}}\right)-F_{1}\left(k_{1_{n}},k_{2_{n}}\right)}{\in }$$
(24)$$\frac{\partial F_{2}}{\partial k_{2}}=\frac{F_{1}\left(k_{1_{n}},k_{2_{n}}+\in \right)-F_{1}\left(k_{1_{n}},k_{2_{n}}\right)}{\in }$$
(25)$$\frac{\partial F_{2}}{\partial k_{1}}=\frac{F_{1}\left(k_{1_{n}}+\in ,k_{2_{n}}\right)-F_{1}\left(k_{1_{n}},k_{2_{n}}\right)}{\in }$$

The Runge–Kutta method numerically solved the simultaneous differential equations of F_1_ and F_2_. Equations (7) and (16) can be symbolized by dC_A_/dt = f1 (t, C_A_, and C_B_) and dC_B_/dt = f2 (t, C_A_, and C_B_) with a known value of k_1_ and k_2_. The values of C_A_ and C_B_ were iterations calculated by increasing t from t = 0 to t = N. Higher incremental values number will increase their accuracy. In this calculation, number N was set to 10,000. The initial iteration was t_0_, then t_1_ = t_0_ + Δt, t_2_ = t_1_ + Δt, until t_n+1_ = t_n_ + Δt. The C_A_ and C_B_ at t = 0 were C_Ao_ and C_Bo_. The next C_A_ and C_B_ were calculated by intermediate constants l_1_, l_2_, l_3_, and l_4_ for function F_1_ and intermediate constants m_1_, m_2_, m_3_, and m_4_ for function F_2_ following Equations (26)–(35). The calculation of C_A_ and C_B_ were iterated until the t = t_N_. Initially, t was set at 0–10 min; then, this calculation was repeated, where the t was set at 20, 30, 40, …, until 100. Thus, the calculated C_A_ and C_B_ values at t = 0, 10, 20, … until 100 were recorded as C_Ai_ and C_Bi_. The value of C_Bi_ contributed to calculating the SSE.
(26)$$l_{1}=f1\left(t_{n},\;C_{A_{n}},C_{B_{n}}\right)\Delta t$$
(27)$$m_{1}=f2\left(t_{n},\;C_{A_{n}},C_{B_{n}}\right)\Delta t$$
(28)$$l_{2}=f1\left(t_{n}+\frac{\Delta t}{2},\;C_{A_{n}}+\frac{k_{1}}{2},C_{B_{n}}+k_{1}/2\right)\Delta t$$
(29)$$m_{2}=f2\left(t_{n}+\frac{\Delta t}{2},\;C_{A_{n}}+\frac{k_{1}}{2},C_{B_{n}}+k_{1}/2\right)\Delta t$$
(30)$$k_{3}=f1\left(t_{n}+\Delta t/2,\;C_{A_{n}}+k_{2}/2,C_{B_{n}}+k_{2}/2\right)\Delta t$$
(31)$$l_{3}=f2\left(t_{n}+\Delta t/2,\;C_{A_{n}}+l_{2}/2,C_{B_{n}}+l_{2}/2\right)\Delta t$$
(32)$$k_{4}=f1\left(t_{n}+\Delta t/2,\;C_{A_{n}}+k_{3}/2,C_{B_{n}}+k_{3}/2\right)\Delta t$$
(33)$$l_{3}=f2\left(t_{n}+\Delta t/2,\;C_{A_{n}}+l_{3}/2,C_{B_{n}}+l_{3}/2\right)\Delta t$$
(34)$$k_{4}=f1\left(t_{n}+\Delta t,\;C_{A_{n}}+k_{3},C_{B_{n}}+k_{3}\right)\Delta t$$
(35)$$l_{4}=f2\left(t_{n}+\Delta t,\;C_{A_{n}}+l_{3},C_{B_{n}}+l_{3}\right)\Delta t$$
(36)$$C_{A_{n+1}}=C_{A_{n}}+\left(k_{1}+k_{2}+k_{3}+k_{4}\right)/6$$
(37)$$C_{B_{n+1}}=C_{B_{n}}+\left(k_{1}+k_{2}+k_{3}+k_{4}\right)/6$$

In this calculation, the error tolerance was set to tol = 0.00001, ε = 0.000001, Δt = 0.001, and N = 10,000. Initially, the isocyanate and hydroxyl concentrations were C_Ao_ = C_Bo_ = 5.43 moles L^−1^. The initially estimated k_1_ and k_2_ were 0.000001, following the reaction rate constant of polyurethane formation at bulk and solution conditions [15,20].

From the calculation, the reaction rate constants of k_1_ and k_2_ were 6.02 × 10^−4^ and 3.08 × 10^−4^ L mole^−1^·min^−1^. Overall, the k_1_, k_2_, and k_3_ values were 6.02 × 10^−4^, 3.08 × 10^−4^, and 9.11 × 10^−4^ L mole^−1^·min^−1^, respectively. 

The value of k_2_ << k_1_ means the branch bonding formation rate was lower than the linear bonding one. The value of k_1_ represents the rate constant of a simpler reaction approach of HTPB and TDI as previously reported by Ajithkumar et al. [19] and Wibowo et al. [21,47] in which the polymerization rate constant in a solution at ambient temperature with the initial mole ratio of NCO to OH (RNCO/OH) 1:1 is 6.0 × 10^−4^ L mole^−1^·min^−1^. This reaction rate is faster than in the bulk system reported by Wibowo [21], Lucio et al. [48], and Olejnik et al. [49]. The value k_3_ >> k_1_ >> k_2_ represents that the diazene oxide bonding formation rate is faster than the urethane and allophanate bonding formation. Generally, the reactivity of nitro groups is higher than hydroxyl or urethane groups. The negative partial charge of the O atom of nitro groups is higher than that of the O atom of hydroxyl groups; thus, nitro groups are more reactive than hydroxyl groups. The coating reaction of hexogen with polyurethane is faster than copolymerization of HTPB and TDI. Initially, each isocyanate groups of TDI react with the nitro group of hexogen; then, the other isocyanate groups of TDI react with hydroxyl to produce urethane. 

### 3.4. Effect of the Reaction Temperature 

The effect of temperature on the reaction rate constant was presented with the Arrhenius relation, shown in Equation (14), where k_i_, A_i_, E_i_, R, and T are the reaction rate constant in reaction step-i (i = 1, 2, 3), collision factor in reaction step i, activation energy in reaction step i, ideal gas constant, and reaction absolute temperature, respectively.
k_i_ = A_i_e^(−Ei/RT)^(38)

The reaction temperature significantly affects the coating and polymerization rate following the Arrhenius equation as presented in Equation (14) [30]. After the reaction rate constant (k_i_) was calculated, ln (k_i_) was plotted versus (1/T), and the linear regression was analyzed as seen in Figure 8. The graphic intercept was ln (A), and the slope was (−E_i_/R). The calculated activation energy of reaction rate constants E_1_, E_2_, and E_3_ were 1178 kJ mol^−1^, 1021 kJ mol^−1^, and 912 kJ mol^−1^. The regression coefficients (*R*^2^ value) were 0.9818, 0.9678, and 0.9560. The effect of the reaction temperature was more significant for a branch bonding than a linear bonding formation, although it had a lower reaction rate constant following E_1_ > E_2_ > E_3_. The first reaction was more sensitive to the change in the reaction temperature. The diazene oxide formation was faster and more sensitive to the reaction temperature than the polyurethane formation. The linear bonding formation was more sensitive to the rising temperature than the branch bonding one. The lower value of activation energy supports this phenomenon. This value was similar to the polymerization of HTPB–TDI at the bulk system reported earlier with the value of activation energy 1152 and 1001 kJ mol^−1^ [25,28]. 

### 3.5. Kinetics Evaluation of IPDI 

Coating kinetics of hexogen and polyurethane-based HTPB was evaluated using IPDI as a curing agent. The isocyanate and diazene oxide concentrations equivalent to the absorbance spectra at 2276 and 1758 cm^−1^ were plotted to time using a calibration curve of isocyanate and diazene oxide concentrations as illustrated in Figure 4 and Figure 5 [20]. The decreasing absorbance meant that the hexogen and HTPB reacted with IPDI. The decreasing isocyanate concentration and the increasing diazene oxide concentration every ten minutes is presented in Figure 9. The decreasing isocyanate concentration followed by the increasing diazene oxide shows that the hexogen was coated with polyurethane. In this case, the running reaction significantly reduced the isocyanate concentration and increased the diazene oxide concentration. The reaction proceeded very slowly for the first 60 min and tended to be constant. The profile of the decreasing isocyanate concentration followed a polynomial-like shape because of the differential Equations (7)–(10).

The reaction rate constants, k_1_ and k_3_, were calculated numerically with the Newton–Raphson method, and the simultaneous differential Equations (7) and (16) were solved numerically using the Runge–Kutta method, as expressed in Equation (20)–(27). The kinetic model was evaluated by solving the simultaneous equations in Equations (7)–(10), similar to the TDI process with a tolerance error of 0.0001. The reaction rate constant, k_3_, was measured by plotting ln C_G_ vs. t as illustrated in Figure 10. The calculated k_3_ is the slope of this graphic, which was 10.04 × 10^−4^ L mole^−1^·min^−1^.

The calculated reaction rate constants k_1_, k_2_, and k_3_ were 7.58 × 10^−4^, 6.89 × 10^−4^, and 10.04 × 10^−4^ L mol^−1^ min^−1^. The reaction rate constant (k_i_) was calculated, ln (k_i_) was plotted versus (1/T), and the linear regression was analyzed as seen in Figure 11. The graphic slope was (−E_i_/R), the activation energy E_1_, E_2_, and E_3_ were 1008, 959, and 724 kJ mol^−1^. The regression coefficients (*R*^2^ value) were 0.9909, 0.9848, and 0.9936. The reaction rate of IPDI was slightly faster than that of TDI. This finding is similar to the kinetic investigation by Zhang et al. and Kamran et al. [29,31] in separate research. The reactivity of isocyanate groups in TDI was more stable than that in IPDI because of the resonance between CH_2_ groups in the aromatic benzene structure [50,51]. This phenomenon resembles polyurethane polymerization from HTPB with several isocyanates in the extrusion process [50]. 

Generally, the kinetic value of hexogen coating with IPDI was similar to that of hexogen with TDI. The branch bonding formation rate was lower than the linear bonding one, and the coating reaction of hexogen was faster than their copolymerization. Therefore, all urethane formed reacted constantly with hexogen to produce a diazene oxide bond. 

## 4. Conclusions

Hexogen coating with polyurethane-based HTPB can be identified from the growth in particle size and sphericity. The coating can be observed from decreasing isocyanate and increasing diazene oxide concentrations using infrared spectroscopy following their absorption at 2276 and 1758 cm^−1^. Moreover, the quantitative analysis was conducted by plotting the absorbance of a particular wavelength on the calibration curve. The evaluation of the kinetics model consisted of several steps: (1) a hydroxyl group and an isocyanate group to produce a urethane group, (2) a urethane group and an isocyanate group to produce an allophanate group, and (3) a nitro group and an isocyanate group to produce a diazene oxide group and carbon dioxide. The constant reaction rate, k_3_, was calculated based on the increasing diazene oxide concentration. The reaction rate constants, k_1_ and k_2_, were numerically calculated using the Newton–Raphson and Runge–Kutta methods based on the decreasing isocyanate concentration. Calculation of the reaction rate constants using the kinetics model developed at an initial mole ratio of isocyanate to hydroxyl group 1:1 generated k_1_, k_2_, and k_3_ values of 6.02 × 10^−4^, 3.08 × 10^−4^, and 9.11 × 10^−4^ L mol^−1^ min^−1^, respectively. In addition, the activation energy of those steps was 1178, 1021, and 912 kJ mol^−1^. There was a slightly different reaction rate constant of hexogen coated with IPDI and TDI. The reaction rate of hexogen coated with IPDI was faster than that coated with TDI. The reaction rate constants k_1_, k_2_, and k_3_ were 7.58 × 10^−4^, 6.89 × 10^−4^, and 10.04 × 10^−4^ L mol^−1^ min^−1^. Meanwhile, the activation energy E_1_, E_2_, and E_3_ were 1008, 959, and 724 kJ mol^−1^.

## Figures and Tables

**Figure 1 polymers-14-01184-f001:**
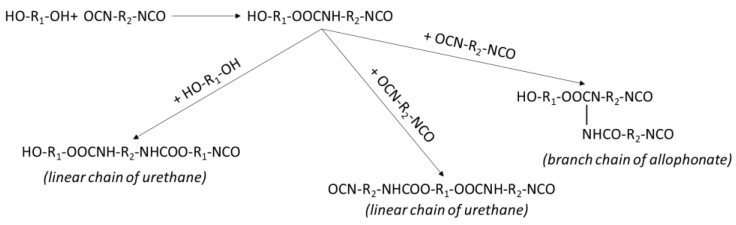
The copolymerization mechanism of diol (HTPB) and diisocyanate (TDI).

**Figure 2 polymers-14-01184-f002:**
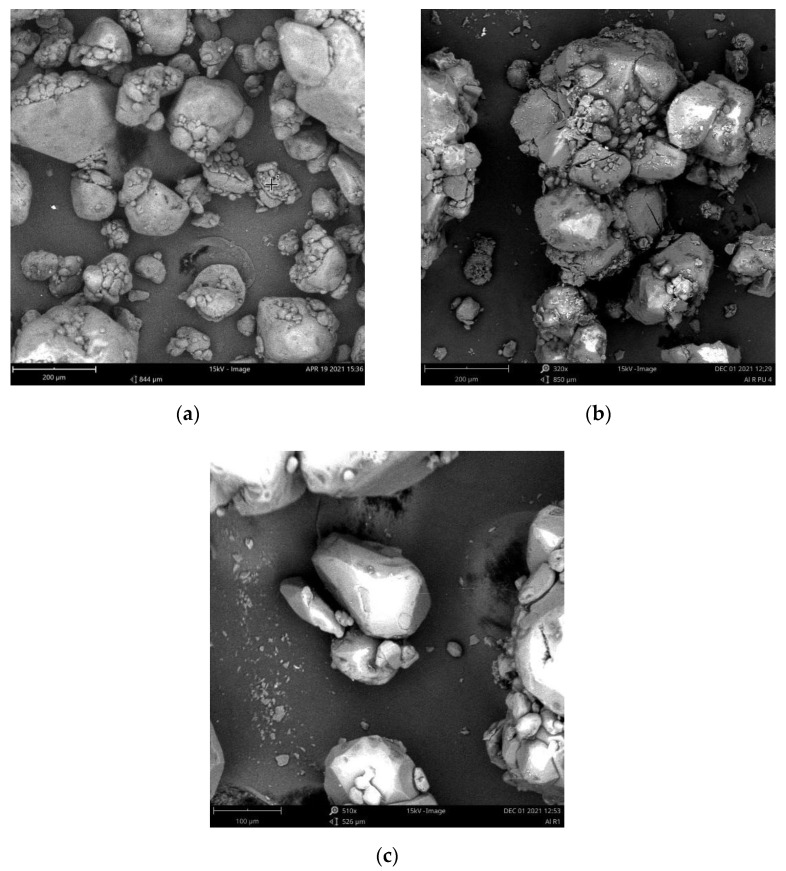
SEM analysis of hexogen (**a**) and coated hexogen (**b**,**c**).

**Figure 3 polymers-14-01184-f003:**
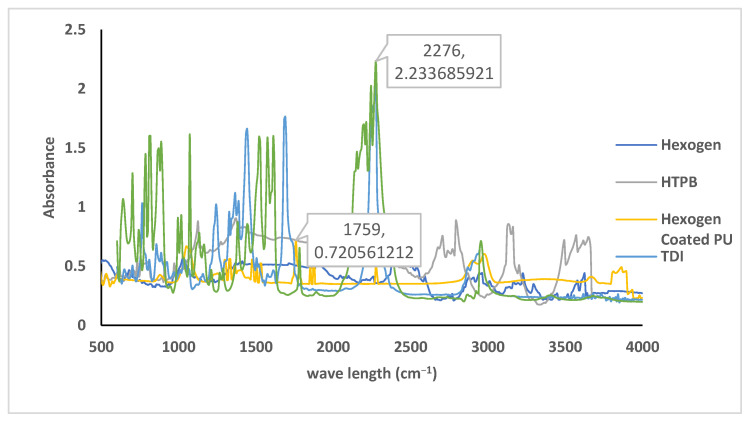
The spectra of ethyl acetate, HTPB, TDI, hexogen, and coated hexogen.

**Figure 4 polymers-14-01184-f004:**
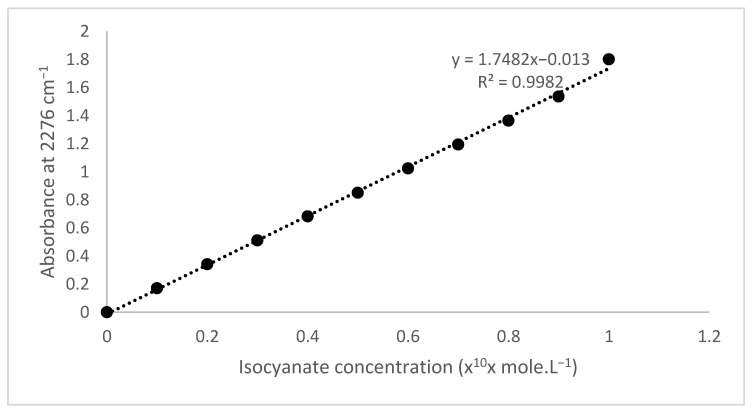
The isocyanate calibration curve based on TDI spectra.

**Figure 5 polymers-14-01184-f005:**
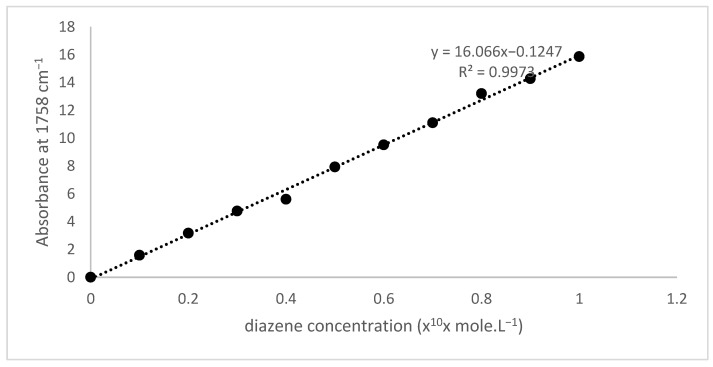
The diazene calibration curve based on 4,4′-dimethoxyazoxybenzene spectra.

**Figure 6 polymers-14-01184-f006:**
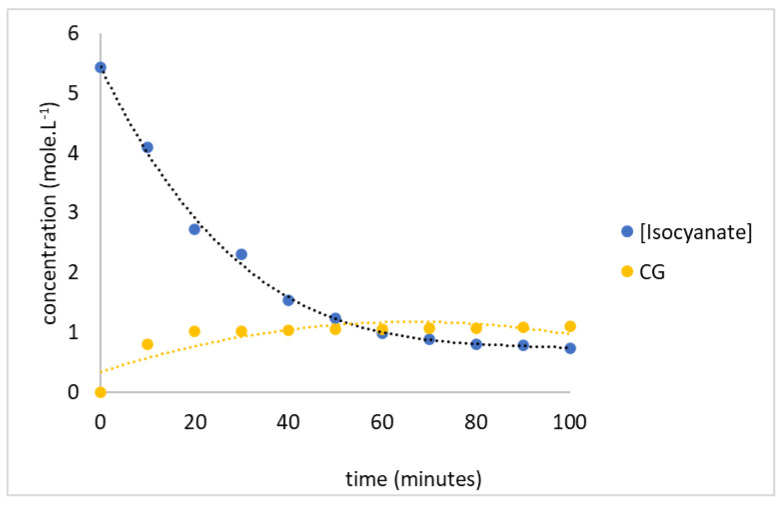
The decreasing isocyanate concentration and increasing diazene oxide concentration following hexogen coating with TDI.

**Figure 7 polymers-14-01184-f007:**
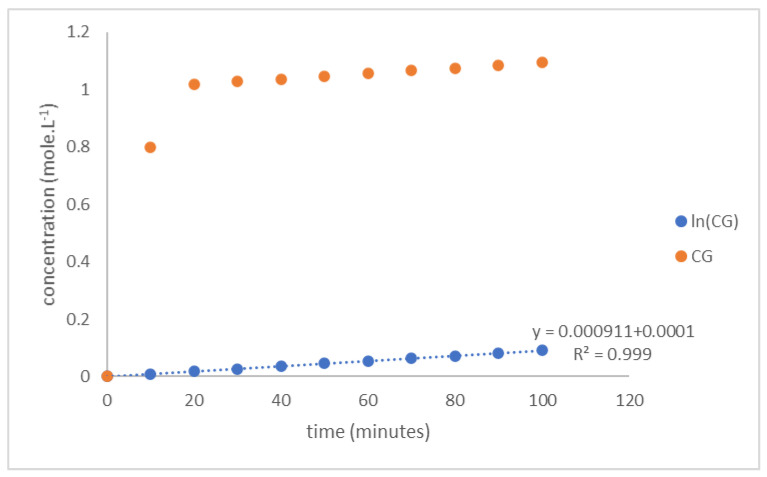
Curve ln C_G_ versus t of hexogen coating with TDI.

**Figure 8 polymers-14-01184-f008:**
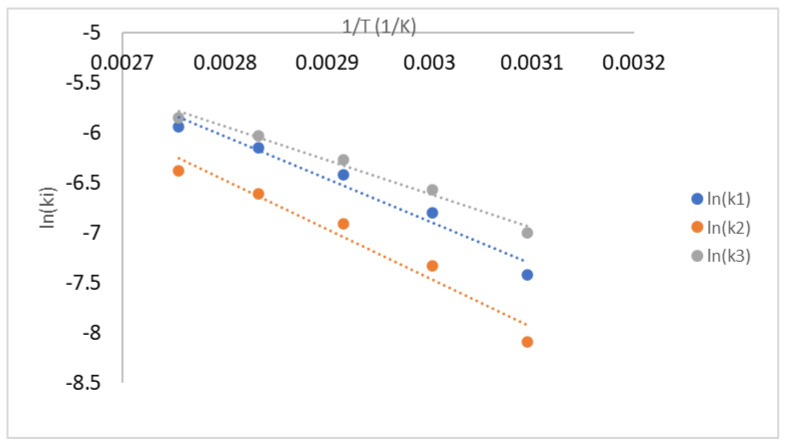
The plot of ln (k_i_) vs. 1/T for hexogen coating with polyurethane.

**Figure 9 polymers-14-01184-f009:**
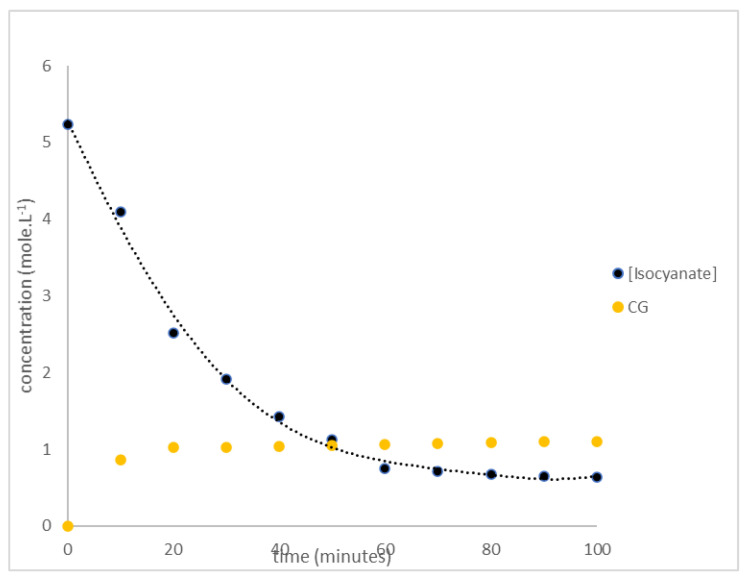
The decreasing isocyanate concentration and increasing diazene oxide concentration following hexogen coating with IPDI.

**Figure 10 polymers-14-01184-f010:**
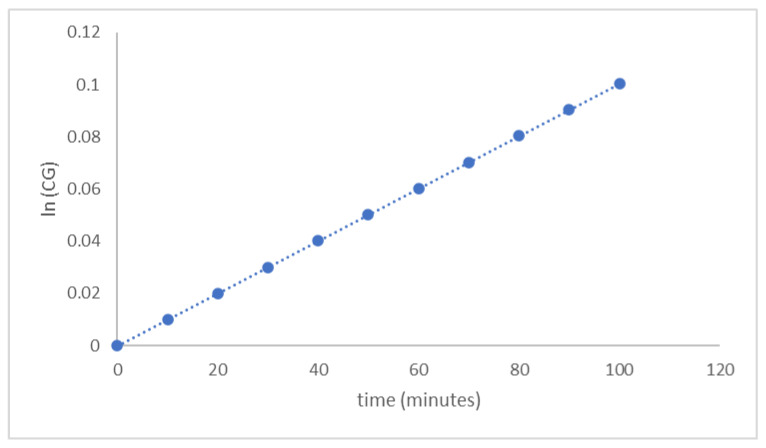
The curve ln C_G_ vs. t of hexogen coating with IPDI.

**Figure 11 polymers-14-01184-f011:**
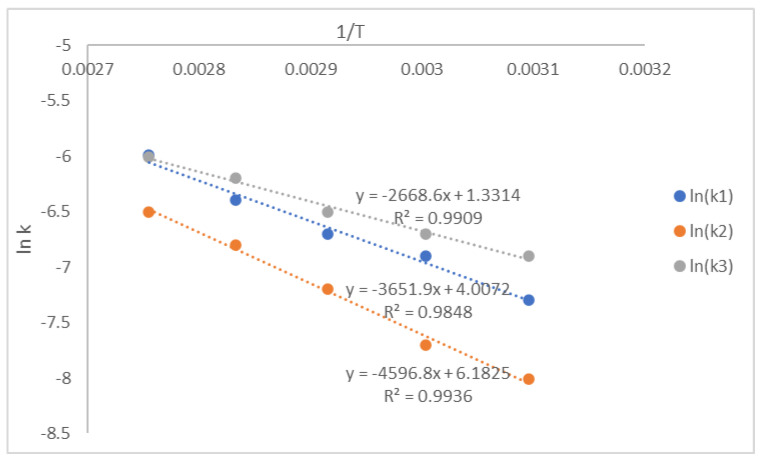
The plot of ln (k_i_) versus 1/T for hexogen coating with polyurethane-based IPDI.

**Table 1 polymers-14-01184-t001:** Comparison between uncoated and coated hexogen particles.

Hexogen	Size/μm	Circularity	Aspect Ratio	Roundness	Solidity
Uncoated (a)	127.68	0.78	1.47	0.70	0.97
PU-Coated IPDI (b)	96.85	0.87	1.30	0.78	0.98
PU-Coated TDI (c)	64.08	0.78	1.36	0.67	0.98

**Table 2 polymers-14-01184-t002:** The calculated and data C_B_ every ten minutes.

t (min)	Calculated C_B_	Data C_B_
0	0.000543	0.000543
10	0.000410	0.000409
20	0.000273	0.000272
30	0.000203	0.000203
40	0.000154	0.000154
50	0.000098	0.000097
60	0.000088	0.000088
70	0.000081	0.000081
80	0.000078	0.000077
90	0.000074	0.000074
100	0.000073	0.000073

## Data Availability

The data presented in this study are available upon request from the corresponding author.

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
