# Peer review of "Hexogen Coating Kinetics with Polyurethane-Based Hydroxyl-Terminated Polybutadiene (HTPB) Using Infrared Spectroscopy"

_polymers, 2022, doi:10.3390/polym14061184_

Round 1
Reviewer 1 Report
This manuscript investigated the kinetics of hexogen coating with PU-based HTPB.
Most of the requests from the reviewers are revised appropriated.
Overall, the current manuscript is recommended for approval to be published in Polymers
Reviewer 2 Report
The Manuscript has been improved to significant extent, It is acceptable for publication
Reviewer 3 Report
Manuscript polymers-1624300 studies the reaction kinetics of hexogen coating with polyurethane-based Hydroxyl Terminated Polybutadiene using infrared spectrometry, and it contains new and significant information to justify publication in Polymers journal, Special Issue "State-of-the-Art Polymeric Surfaces and Coatings.
The article is presented in a well-structured manner, and the methods / procedures used, the results obtained and their interpretation are clearly presented and I consider that they do not require modifications or corrections.
Even if only about 30% of the reference are from the last 5 years and there are 6 self-citations, the references are adequate to the content.
There are typos that will be corrected
I propose to accept the manuscript in the current form.
This manuscript is a resubmission of an earlier submission. The following is a list of the peer review reports and author responses from that submission.
Round 1
Reviewer 1 Report
In this manuscript, the authors studied the kinetics of hexogen coating on HTPB-based polyurethane by infrared spectrometry. The quality of the manuscript needs to be enhanced before publication.
- The language and grammar of the manuscript should be thoroughly washed. It is not reader-friendly at this moment.
- Unlike NMR, IR is considered as a qualitative technique, which means that the transmission intensity might not be accurate enough to demonstrate the reaction progress.
- More characterizations should be performed to support the work. The manuscript is not solid enough.
Reviewer 2 Report
The article at hand by Wibowo et al reports on the kinetics of three reactions happening simultaneously, during coating of hexogen particles with a common polyurethane. This is done on the basis of monitoring the concentration of isocyanate (which is consumed in all reactions) and the diazene oxide (which is produced by one of them). By Runge Kutta methods the authors calculate the reaction constants. The topic of the work and the approach followed by the authors is of some interest, but the implementation has several shorcomings that do not allow me to suggest this article for publication in Polymers.
MAJOR ISSUES
- How was FTIR data converted to concentration? Did the authors measure a callibration curve? Is there at least evidence that intensity is linear with concentration?
- Could the authors describe exactly the method used for the determination of the three reaction constants? What are the inaccuracies (i.e. the errorbars) in determining the raw data (concentration) and the three reaction constants? It seems far fetched to me that with 2 curves of 7 points each, three reaction constants can be calculated with any reasonable accuracy.
- The data for IPDI should also be shown to some extent.
minor issues
- acronym of HTPB must be defined. I assumed Hydroxyl-terminated polybutadiene
- line 158. What is the difference between circularity and roundness?
- Figure 5 has too much white space